# Maternal Vegetable and Fruit Consumption during Pregnancy and Its Effects on Infant Gut Microbiome

**DOI:** 10.3390/nu13051559

**Published:** 2021-05-05

**Authors:** Hsien-Yu Fan, Yu-Tang Tung, Yu-Chen S. H. Yang, Justin BoKai Hsu, Cheng-Yang Lee, Tzu-Hao Chang, Emily Chia-Yu Su, Rong-Hong Hsieh, Yang-Ching Chen

**Affiliations:** 1Department of Family Medicine, Taipei Medical University Hospital, Taipei 110, Taiwan; d07849004@ntu.edu.tw; 2Institute of Epidemiology and Preventive Medicine, National Taiwan University, Taipei 100, Taiwan; 3Graduate Institute of Metabolism and Obesity Sciences, College of Nutrition, Taipei Medical University, Taipei 110, Taiwan; peggytung@nchu.edu.tw; 4Graduate Institute of Biotechnology, National Chung Hsing University, Taichung 402, Taiwan; 5Joint Biobank, Office of Human Research, Taipei Medical University, Taipei 110, Taiwan; can_0131@tmu.edu.tw; 6Department of Medical Research, Taipei Medical University Hospital, Taipei 110, Taiwan; justin.bokai@gmail.com; 7Office of Information Technology, Taipei Medical University, Taipei 110, Taiwan; nathanlee@tmu.edu.tw (C.-Y.L.); kevinchang@tmu.edu.tw (T.-H.C.); 8Graduate Institute of Biomedical Informatics, College of Medical Science and Technology, Taipei Medical University, Taipei 110, Taiwan; emilysu@tmu.edu.tw; 9Clinical Big Data Research Center, Taipei Medical University Hospital, Taipei 110, Taiwan; 10School of Nutrition and Health Sciences, College of Nutrition, Taipei Medical University, Taipei 110, Taiwan; hsiehrh@tmu.edu.tw; 11Department of Family Medicine, School of Medicine, College of Medicine, Taipei Medical University, Taipei 110, Taiwan

**Keywords:** nutrients, infant gut microbiome, pregnancy, vegetables, fruits

## Abstract

Maternal nutrition intake during pregnancy may affect the mother-to-child transmission of bacteria, resulting in gut microflora changes in the offspring, with long-term health consequences in later life. Longitudinal human studies are lacking, as only a small amount of studies showing the effect of nutrition intake during pregnancy on the gut microbiome of infants have been performed, and these studies have been mainly conducted on animals. This pilot study explores the effects of high or low fruit and vegetable gestational intake on the infant microbiome. We enrolled pregnant women with a complete 3-day dietary record and received postpartum follow-up. The 16S rRNA gene sequence was used to characterize the infant gut microbiome at 2 months (*n* = 39). Principal coordinate analysis ordination revealed that the infant gut microbiome clustered differently for high and low maternal fruit and vegetable consumption (*p* < 0.001). The linear discriminant analysis effect size and feature selection identified 6 and 17 taxa from both the high and low fruit and vegetable consumption groups. Among the 23 abundant taxa, we observed that six maternal intake nutrients were associated with nine taxa (e.g., *Erysipelatoclostridium*, *Isobaculum*, Lachnospiraceae, Betaproteobacteria, Burkholderiaceae, *Sutterella*, Clostridia, Clostridiales, and *Lachnoclostridium*). The amount of gestational fruit and vegetable consumption is associated with distinct changes in the infant gut microbiome at 2 months of age. Therefore, strategies involving increased fruit and vegetable consumption during pregnancy should be employed for modifying the gut microbiome early in life.

## 1. Introduction

According to the Development Origins of Health and Disease (DOHaD) hypothesis, maternal nutrition in pregnancy has a significant impact on offspring disease risk in the future [1]. A maternal diet rich in fruits and vegetables during pregnancy is associated with a reduced risk of allergic diseases [2], and an increased risk of obesity [3]. Given that nutrient intake strongly influences microbiome function and relative abundance, the infant gut microbiome might be a potential mediating factor linking gestational nutritional exposure and future childhood diseases [4].

Maternal nutrition during pregnancy may affect the mother-to-child transmission of bacteria, resulting in gut microflora changes in the child, with long-term consequences after birth [4]. However, evidence supporting the effect of maternal nutrition during pregnancy on the infant gut microbiome remains scarce, and most relevant studies have been conducted on animal models. Chu et al. discovered that a high-fat maternal diet during gestation shapes the offspring gut microbiome in animals (Japanese Macaque) [5] and humans [6]. The mother’s high-fat diet was shown to damage the microbiome and immune system of their offspring [7]. The offspring of mothers who consume Western diets displayed a significantly increased effect of Pachycephalospora on Bacteroides, and the microbiome of the offspring of mothers who were fed a high-fat diet had an increased ability to extract energy from the diet. Using a sow model, Li et al. reported that maternal dietary fiber intake alters offspring gut microbiome composition [8]. Similarly, maternal fruit intake was associated with an increased risk of a high *Streptococcus*/*Clostridium* gut microbiome composition among vaginally delivered infants [9]. Possible mechanisms for the effects of maternal diet during pregnancy on the infant gut microbiome include the transmission of nutrients through amniotic fluid, vaginal delivery, or the placenta. However, the effect of gestational intake of high or low fruit and vegetable intake on the infant microbiome remains unclear in the study of humans.

Several studies suggest that supplementation with nutrients found in fruits and vegetables, such as dietary fiber, vitamin C, and fructose, could modulate the structure of host gut microbes [10]. According to a previous study by Alison et al., a high-fiber diet alters gut microbial ecology and causes significant perturbations at the phylum level [11]. Li et al. found that vitamin C could strongly modulate the gut microbiota [12]. In another animal study, the maternal diet supplemented with fructose appeared to regulate the maternal microbiome significantly, causing infant gut dysbiosis [13]. The intake of maternal dietary fruits and vegetables may not only have an effect on the host, but also on their offspring [14,15]. However, this relationship has, to date, been poorly understood.

In this study, we (1) explored the impact of high/low gestational intakes of fruits and vegetables on the infant microbiome, and (2) investigated the interrelationships between maternal nutrients and the abundance of infant gut microbiome taxa.

## 2. Materials and Methods

### 2.1. Cohort Establishment and Data Collection

In July 2018, we formed a Taipei Mother–Infant Nutrition Cohort, which was approved by the Joint Institutional Review Board of Taipei Medical University (N201811050). The 1008 pregnant women in the initial cohort were enrolled from three hospitals and four obstetric clinics, between 2018 and 2019, and they all provided informed consent for this study (Appendix A). Pregnant women with severe diseases (e.g., heart diseases) were excluded. Initially, we included 479 participants who completed the baseline survey and provided dietary data, and 199 of them went on to participate in newborn follow-up visits. We excluded participants who were unwilling to provide infant stool samples in this study. By the end of April 2020, 39 infant stool samples were obtained at 2 months postpartum during home visits.

### 2.2. Maternal Dietary Assessment

All participants received assistance in installing an image-based dietary assessment application on their smartphones (Cofit Pro version 1.0.0, Taipei, Taiwan). We previously proved the validity and reliability of this image-based dietary assessment application for assessing macronutrients and micronutrients [16]. Registered dietitians taught the participants to use the application on-site for 20–30 min. Participants used their smartphones to take photos of all the food that they consumed for ≥3 days. Dietary records from three matching days (two weekdays and one weekend day) were used for this analysis. Dietary variables were calculated as individual means of the 3-day dietary records. After food record collection, trained dietitians disaggregated the foods into their constituent ingredients, including macronutrients (measured in grams) and micronutrients (measured in milligrams). The macronutrients included carbohydrates and dietary fiber, fats, proteins, and fluids. The micronutrients included vitamins and minerals.

Fruits and vegetables contribute the most to dietary fiber intake in the Taiwanese population; therefore, the 2018 Dietary Guidelines of Taiwan for pregnant women recommend the consumption of 5–9 cups of fruits and vegetables per day. More than 80% of women in our cohort consumed fewer fruits and vegetables than this recommendation during pregnancy. On average, they consumed 4.9 g/day of dietary fiber, well below the recommended 25 g/day. The mean cups of fruits and vegetables were estimated as follows: five cups of fruits and vegetables (the minimum recommended), multiplied by 20% (derived from the average 4.9 g/day divided by the recommended 25 g/day of dietary fiber). The high or low consumption of vegetables and fruit was determined based on more than one cup of fruits and more than one cup of vegetables per day.

### 2.3. Sample Collection and DNA Extraction

Stool samples were collected during home visits when the infants were 2 months of age. Before home visits, we mailed the participants stool sample collection tubes, which contained commercial deoxyribonucleic acid (DNA) stabilization buffer, to protect DNA from degradation after collection. DNA stabilization contained RNAlater, which protects DNA from degradation at room temperature from days to weeks [17]. The bacterial DNA was extracted using a Qiagen DNA Mini Kit (Qiagen, Hilden, Germany) and stored at −80 °C.

### 2.4. Targeted 16S rRNA Gene Sequencing

The analytical methods of 16S rDNA analysis were established in a previous study [18]. By referencing Illumina’s recommended protocols (https://support.illumina.com/downloads/16s_metagenomic_sequencing_library_preparation.html; accessed on 10 December 2020), we performed library construction and amplification of the 16S rRNA gene. In summary, we used the forward and reverse primers 341F and 805R with Illumina overhang adapter sequencing to amplify the V3–V4 region of the bacterial 16S rRNA gene. A Nextera XT Index kit was then used to adjust the dual-index barcodes to the targets in the amplicon and the Illumina sequencing adapters. The quantity and quality of data in the sequenced library were assessed using a QSep100 analyzer (BiOptic, Taipei, Taiwan). Moreover, by using a MiSeq Reagent kit v3, high-throughput sequencing was performed on an Illumina MiSeq 2000 sequencer.

The bioinformatics analytical process was conducted following the workflow described by Callahan et al. [19]. First, by using the R package DADA2 (v 1.14.1), the filtered reads were managed.Taxonomy assignment was administered using the SILVA database (v128) with a minimum bootstrap confidence level of 80 [20]. Multiple sequence alignment of the structural variants was processed with DECIPHER (v2.14.0), and a phylogenetic tree was built from the alignment using phangorn (v2.5.5) [21]. The count table, taxonomy assignment results, and phylogenetic tree were consolidated into a phyloseq object, and community analyses were created by phyloseq (v1.30.0) [22]. One-way ANOVA followed by the Bonferroni post hoc test were utilized to handle multiple comparison analysis. The analytical process of alpha-diversity and beta-diversity are listed below. The phyloseq package was used to calculate the alpha-diversity. For beta-diversity, principal coordinate analysis (PCoA) was performed on UniFrac distances, and the adonis and betadisper functions from the vegan package (v2.5.6) were used to analyze the dissimilarity of composition among high- and low-consumption groups. The groups were compared with α = 0.05 (Kruskal–Wallis and Wilcoxon tests). The UniFrac package (v1.1) was used to compare the community dissimilarity between groups, demonstrated as UniFrac distances [23]. GraPhlAn [24] helped us to perform the enrichment analysis between the groups, which were analyzed using the linear discriminant analysis (LDA), effect size, (LEfSe) method, and a logarithmic LDA score of more than 2 [25] and were then visualized as a cladogram.

### 2.5. Confounding Factors

We conducted surveys once during pregnancy and twice after birth. The questionnaire administered to the mothers included demographic data for mothers and children, maternal health and disease status, breastfeeding status, perinatal antibiotics use, and children’s health and disease status. Data of potential confounders that would influence the infant gut microbiome, such as delivery mode, gestational age, and gestational weight gain, were also collected.

### 2.6. Data Analysis

Chi-square and t test were used to examine whether demographic characteristics and maternal nutrients differ between the groups. Linear regression was used to examine the relationship between nutrients and the infant gut microbiome at 2 months of age.

## 3. Results

### 3.1. Demographic Characteristics

As presented in Table 1, the percentage intake of vegetables (*p* < 0.001) and fruits (*p* = 0.08) differed significantly between the groups, whereas that of dairy, grain, meat, and fat did not differ significantly between groups (all *p* > 0.05). Other potential confounders, such as gestational age (preterm birth or not), excess gestational weight gain, mode of delivery, and breastfeeding or formula feeding, were also not significantly different (all *p* > 0.05).

### 3.2. Variation of Maternal Nutrient Intake

Maternal nutrient intake (macronutrients and micronutrients) in the high and low vegetable and fruit consumption groups during pregnancy is presented in Figure 1. The mothers with high fruit and vegetable consumption had a significantly higher intake of macronutrients (glucose, fructose, and dietary fiber), vitamins (folic acid and ascorbic acid), and minerals (potassium) than mothers with low fruit and vegetable consumption.

### 3.3. Variation of Infant Gut Microbiome

High or low maternal consumption of vegetables or fruits during pregnancy did not affect the alpha diversity of the infant’s gut microbiome (Appendix A). To establish the effect of maternal fruit and vegetable intake during pregnancy on the infant’s gut microbiome composition, we conducted Illumina-generated 16S rRNA amplicon sequencing from 39 samples. The PCoA based on unweighted UniFrac distances revealed that the microbiome of 2-month-old infants varied depending on whether the maternal consumption of fruits and vegetables gestation was high or low (Figure 2A). However, other potential confounders, such as maternal age, maternal education level, family income, gestational age, excess gestational weight gain, delivery mode, antepartum antibiotics, group B *Streptococcus* positivity, sex of the infant, and breastfeeding did not affect the infant gut microbiome (Appendix A). As shown in Figure 2B, LEfSe revealed that the counts of Propionibacteriales, Propionibacteriaceae, *Cutibacterium*, Tannerellaceae, *Parabacteroides*, and *Lactococcus* were higher in the microbiome of 2-month-old infants with high maternal vegetable and fruit consumption. However, the counts of *Prevotella_2*, *Prevotella_9*, *Isobaculum*, Clostridia, Clostridiales, Lachnospiraceae, *Hungatella*, *Lachnoclostridium*, Ruminococcaceae, *Flavonifractor*, *Erysipelatoclostridium*, Acidaminococcaceae, *Phascolarctobacterium*, *Megamonas*, Betaproteobacteriales, Burkholderiaceae, and Sutterella were higher in the microbiome of 2-months-old infants with low maternal fruit and vegetable consumption.

### 3.4. Maternal Nutrient Intake and Infant Gut Microbiome

Heatmaps of the correlation between maternal nutrient intake during pregnancy and infant gut microbiome at 2 months of age are displayed in Figure 3. A high-fructose maternal diet was negatively associated with *Erysipelatoclostridium*. A high-glucose maternal diet was significantly associated with an enrichment of *Isobaculum* in the infant gut microbiome. The Lachnospiraceae count was lower among infants with higher maternal consumption of dietary fiber (Figure 3A). Betaproteobacteria, Burkholderiaceae, and *Sutterella* were strongly negatively correlated with folic acid, and Betaproteobacteria and Burkholderiaceae were negatively correlated with ascorbic acid (Figure 3B). Clostridia, Clostridiales, and Lachnospiraceae were negatively correlated with both magnesium and potassium, and *Lachnoclostridium* was negatively correlated with potassium.

As shown in Appendix A, counts of *Hungatella* and *Megamonas* were lower among infants with higher maternal consumption of vegetables. Moreover, the count of *Erysipelatoclostridium* was lower among those with higher maternal consumption of fruits, and the count of *Megamonas* was higher among those with higher maternal consumption of dairy. The count of *Isobaculum* was lower among those with higher maternal consumption of grains, whereas the count of *Flavonifractor* was higher. The counts of both Lachnospiraceae and *Lachnoclostridium* were lower among those with higher maternal consumption of meat.

## 4. Discussion

We demonstrated that, in this mother–infant nutrition cohort, the infant gut microbiome at 2 months of age varied according to the level of maternal fruit and vegetable consumption during pregnancy. We identified 6 and 17 taxa in the infant gut microbiome from the high and low fruit and vegetable consumption groups, respectively. Furthermore, we have shown the detailed nutrients and gut microbiome taxonomic interactions.

### 4.1. Maternal Fruit and Vegetable Consumption Affects Infant Microbiome

High or low maternal fruit and vegetable consumption was significantly correlated with infant gut microbiome composition. Gut microbiome composition is related to the intake of dietary fiber [26], which is fermented by certain bacteria, producing short-chain fatty acids (SCFAs) such as acetate, propionate, and butyrate [27]. Animal studies have reported an association of increased maternal dietary microbiome-accessible fiber and SCFA exposure during pregnancy, with a reduced incidence of asthma in offspring [11,28], and this effect persists into adulthood [11]. A follow-up small human component (*n* = 61) of the same study indicated that an association exists between reduced dietary fiber intake and reduced serum acetate levels in pregnant women. A separate component (*n* = 40) revealed a correlation between serum acetate levels that were lower than the median, and increased frequency of coughing/wheezing during the child’s first year of life [11,28].

In a mouse model study, the plasma SCFA levels of the offspring of mice fed a high-fiber diet were higher than those of mice fed a no-fiber diet, and the frequencies of thymic regulatory T cells (Tregs) and peripheral Tregs were higher in the offspring of high-fiber-diet-fed mice [26]. During pregnancy, SCFA (such as acetate) can cross the placenta and affect the expression of fetal lung genes, such as NPPA, which encodes ANP (a molecule related to epithelial biology and immune regulation) [11]. In demonstrating associations between maternal high dietary fiber intake, antenatal exposure to SCFAs, and offspring allergic diseases, these mouse experiments have suggested a possible target for interventions to reduce the burden of allergic diseases; however, no clinical trials have investigated the protective effects of maternal microbiome against allergic diseases through a maternal high-fiber diet.

In the present study, we observed a higher relative abundance of *Cutibacterium*, *Parabacteroides*, and *Lactococcus* in the fecal microbiome of infants exposed to high vegetable and fruit consumption during gestation. However, the higher abundance of *Prevotella_2*, *Prevotella_9*, *Isobaculum*, *Hungatella*, *Lachnoclostridium*, *Flavonifractor*, *Erysipelatoclostridium*, *Phascolarctobacterium*, *Megamonas*, and *Sutterella* was associated with low fruit and vegetable consumption. Our findings regarding the beneficial effects of *Cutibacterium*, *Parabacteroides*, and *Lactococcus* on infant immunity are consistent with those of previous studies. *Cutibacterium* ferments hexoses through the Embden–Meyerhof pathway to produce pyruvate, which is further metabolized into propionate [29], whose consumption was reported to reduce antigen presentation on dendritic cells as a result of GPR41-dependent modulation of hematopoiesis and affected allergic diseases in a mouse model [28]. *Parabacteroides* have multiple beneficial effects on human health. P. distasonis can improve human bowel health [30] and is negatively associated with celiac disease [31]. It can reduce weight gain, hyperglycemia, and liver steatosis in ob/ob and high-fat diet mice [32] and significantly reduce the severity of intestinal inflammation in murine models of acute and chronic colitis [33]. The SCFA-producing *Parabacteroides* were richer in the cecum and colorectum, where, accordingly, more SCFAs were produced [34]. *Lactococcus* lactis activates innate immunity and protects from infections [35,36]. Moreover, some Lactobacilli can produce SCFAs [34], which can induce Tregs to modulate gut immune responses [37,38,39], and can shape the pulmonary immune environment and influence the severity of allergic inflammation [28].

### 4.2. Maternal Difference in Six Nutrients and 23 Bacterial Taxa

A comparative study on fecal samples from volunteers with diets low in fructose—a determinant of microbial diversity—revealed that the relative abundance of *Erysipelatoclostridium* was lower among those with a high-fructose syrup diet than among those on a fruit-based diet [40]. The effects of maternal nutrients on the infant gut microbiome have never before been examined in a human model. Here, we highlight that maternal exposure to fructose reduces the abundance of *Erysipelatoclostridium* in the infant gut microbiome. Previous studies on rats have demonstrated that fructose adversely affects intestinal permeability and disrupts the maternal microbiome, leading to altered offspring gut development [13,41]. Fructose may inhibit the growth of harmful flora and promote the growth of beneficial and neutral flora.

Micronutrients may be associated with the abundance of certain taxa in the infant gut microbiome. For example, higher consumption of folate is associated with a lower abundance of Lachnospiraceae [42]. Folate explains 8% of the relative abundance of Lachnospiraceae [42]. Here, we have demonstrated that folate was significantly inversely associated with the abundance of Betaproteobacteria. In the present study, the abundance of Lachnospiraceae was inversely affected by dietary fiber, magnesium, and potassium. However, the effects of micronutrients in the above association in the mother or child remain unclear. Regarding maternal vitamin intake and gut microbiome, the intake of ascorbic acid (vitamin C) during pregnancy was positively correlated with the abundance of *Staphylococcus* [43]. Although the role of ascorbic acid in *Staphylococcus* metabolism remains unclear, both have been linked to the immune profile [44].

### 4.3. Strengths and Limitations

This is the first human study to demonstrate that low gestational consumption of fruits and vegetables affects the infant gut microbiome. Our data support previous findings from animal studies [11]. Moreover, instead of using a food frequency questionnaire, we used 3-day dietary records to obtain details regarding nutritional intake, thus enabling us to investigate the correlations between gestational nutrition and the abundance of infant gut microbiome taxa.

Our study has some limitations. The generalizability of our findings may be limited by our relatively small sample size. However, the identification of three unhealthy infant gut microbial taxa in the low vegetable/fruit consumption group agrees with the data obtained from other studies conducted in Asian countries [45,46]. Moreover, we were unable to collect samples for evaluating the infant gut microbiome and SCFA at multiple time points. Larger and longer studies that better account for antenatal and postnatal nutritional exposure factors are warranted, to elucidate the detailed mechanisms linking gestational nutritional exposure to early allergic diseases or other chronic diseases.

## 5. Conclusions

A maternal diet rich in fruits and vegetables during pregnancy may alter the infant gut microbiome. Higher maternal nutritional intake of fructose, dietary fiber, folic acid, and ascorbic acid was negatively associated with the abundance of unhealthy infant gut microbiomes, such as *Erysipelatoclostridium*, Betaproteobacteria, and Lachnospiraceae. Therefore, strategies should be applied for modifying the gut microbiome early in life through the promotion of a higher intake of fruits and vegetables during pregnancy.

## Figures and Tables

**Figure 1 nutrients-13-01559-f001:**
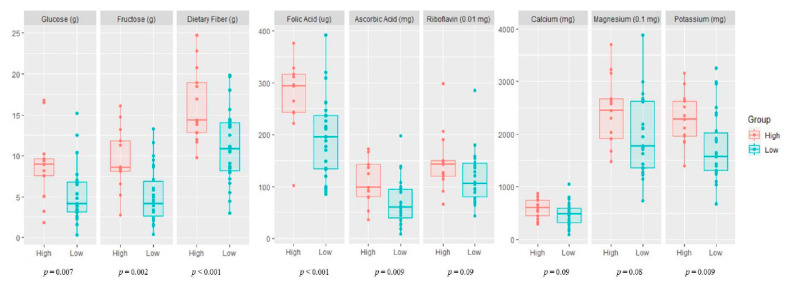
Maternal nutrient intake during pregnancy according to vegetable and fruit consumption. High, high maternal consumption of fruits and vegetables during pregnancy; Low, low maternal consumption of fruits and vegetables during pregnancy.

**Figure 2 nutrients-13-01559-f002:**
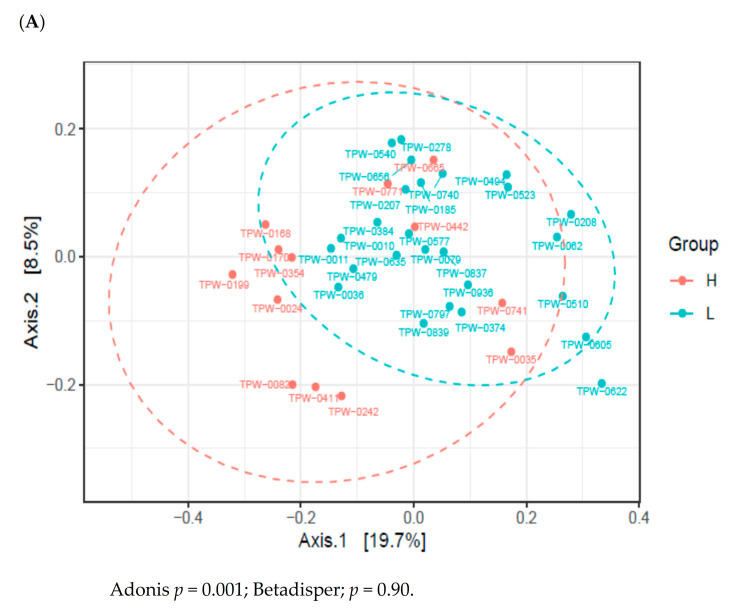
Variations in the infant gut microbiome at 2 months of age according to maternal fruit and vegetable consumption during pregnancy. (**A**) Principal coordinates analysis (PCoA) on unweighted unique fraction (UniFrac). (**B**) linear discriminant analysis effect size (LEfSe). The ordination is from A to W in a tree diagram (Appendix A). Definition of abbreviation: H, high maternal consumption of fruits and vegetables during pregnancy; L, low maternal consumption of fruits and vegetables during pregnancy; O, order; F, family; G, genus.

**Figure 3 nutrients-13-01559-f003:**
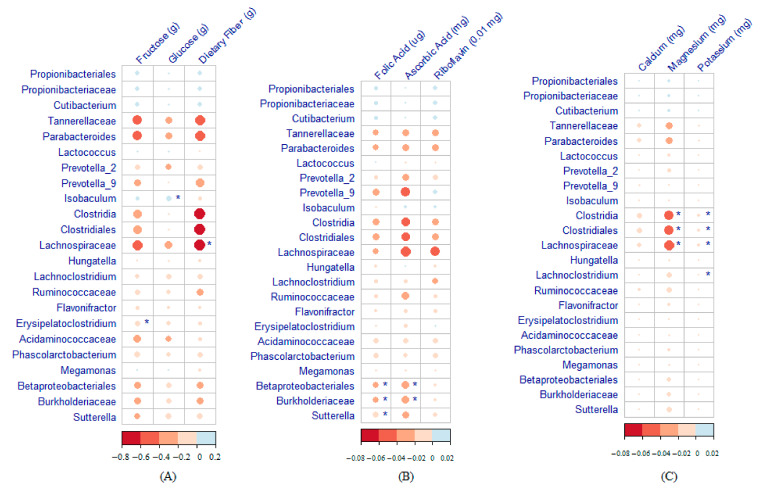
Heatmaps showing the correlation between maternal nutrient intake during pregnancy and the infant gut microbiome at 2 months of age (* Significant association). (**A**) Macronutrients. (**B**) Vitamins. (**C**) Minerals.

**Table 1 nutrients-13-01559-t001:** Characteristics of groups with high and low maternal consumption of fruits and vegetables during pregnancy.

Characteristics	High Consumption * (*n* = 13)		Low Consumption (*n* = 26)		Comparison	
	N or Mean	% or (SD)	N or Mean	% or (SD)	Statistics	*p*
Maternal age at baseline	34.2	(2.6)	33.5	(4.5)	0.64	0.53
<30	0	0.0%	5	19.2%	2.87	0.32
30–35	7	53.8%	11	42.3%		
≥35	6	46.2%	10	38.5%		
Maternal education level						
Senior high school or below	0	0.0%	1	3.8%	0.67	0.82
College	8	61.5%	17	65.4%		
Graduate school and higher	5	38.5%	8	30.8%		
Family income						
<60,000	5	38.4%	10	38.4%	2.14	0.32
60,000–100,000	5	38.4%	5	19.2%		
>100,000	3	23.1%	11	42.3%		
Maternal history of diseases						
Cardiovascular disease	1	7.6%	0	0.0%	0.12	0.72
Gestational diabetes mellitus	0	0.0%	1	3.8%	0.00	1.00
Hyperthyroidism	2	15.3%	0	0.0%	1.64	0.20
Hypothyroidism	1	7.6%	0	0.0%	0.12	0.72
Timing of dietary assessment						
<13 weeks	0	0.0%	8	30.8%	5.12	0.08
13–28 weeks	8	61.5%	12	46.2%		
>28 weeks	5	38.5%	6	23.1%		
Gestational age	38.7	(1.3)	38.1	(1.2)	1.20	0.24
<37 weeks	1	8.3%	2	8.3%	0.00	1.00
Excess gestational weight gain	2	15.4%	4	15.4%	0.00	1.00
Normal spontaneous delivery	9	69.2%	17	65.4%	0.00	1.00
Antepartum antibiotics	1	7.7%	4	15.4%	0.03	0.64
Group B streptococcus positive	2	15.4%	5	19.2%	0.00	1.00
Neonatal sex (male)	7	58.3%	17	65.4%	0.04	0.72
Breastfeeding (yes)	5	38.5%	10	38.5%	0.00	1.00
Dietary intake						
Calories (kcal)	1679.3	(354.4)	1541.1	(378.9)	204.0	0.31
Vegetables (cups ^†^)	2.1	(0.7)	1.8	(1.5)	228.0	0.08
Fruit (cups ^†^)	1.9	(0.5)	0.6	(0.6)	317.0	<0.001
Dairy (cups ^†^)	0.7	(0.5)	0.6	(0.8)	202.5	0.32
Grain (1/4 cups ^†^)	8.4	(2.1)	8.2	(2.4)	175.0	0.87
Meat (1/6 cups ^†^)	5.9	(2.4)	6.0	(3.3)	185.0	0.65
Fat (tbsp ^†^)	5.5	(1.8)	5.9	(2.3)	150.0	0.59

* High consumption was defined as ≥1 cup of fruits or vegetables per day. ^†^ One cup = 240 mL; tbsp = tablespoon (5 mL).

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
