# Peer review of "Maternal Vegetable and Fruit Consumption during Pregnancy and Its Effects on Infant Gut Microbiome"

_nutrients, 2021, doi:10.3390/nu13051559_

Round 1
Reviewer 1 Report
Thank you for the paper. I have a few comments.
1) When was the mothers dietary intake measured? During pregnancy or after?
2) Was antibiotic intake associated with gut microbiome?
3) Did maternal history of allergy affect any of your exposure or outcome data?
4) Why did you ask about allergic rhinitis – very difficult to measure at one year.
5) What about food allergy – do you have any data?
6) You found no effect of diversity, the main driver of butyrate production. Do you think other fibers may be more important than fruit and vegetables to drive diversity?
7) Why would the 3 strains you found be association with atopic dermatitis outcomes? Do they have a specific role in the gut microbiome? Have others found these strains are important in atopic dermatitis prevention?
Author Response
Dear Editor,
Enclosed please find our revised manuscript. We thank you for allowing us the opportunity to revise our manuscript. Our responses to the reviewers’ comments are attached in the following pages. We have carefully answered all concerns, and some alterations (highlighted in red font) have been made according to the reviewers’ suggestions. We have rewrite the introduction and methodology section to address the reviewer’s concern. We really appreciate the helpful comments and hope that our manuscript is now acceptable for publication in your prestigious journal.
With very best wishes and warm regards.
Response to Reviewer # 1 comments
Comment 1: When was the mothers dietary intake measured? During pregnancy or after?
Response 1: Dietary intake were assessed for 2nd or 3rd trimester (mostly)- pregnant mothers in Taipei Mother–Infant Nutrition Cohort. In Table R1, we listed the macronutrients intake of mothers recruiting from three trimesters and it did not differ much. We then modified the section “method” to make the description more clear in the 1st paragraph of Materials and Method. The revised description was listed as below, “The 1,008 pregnant women in the initial cohort were enrolled from three hospitals and four obstetric clinics between 2018 and 2019 and provided informed consent (Figure S1). We included 479 pregnant women who completed the baseline survey and provided dietary data, and 199 of them participated in newborn follow-up visits.”
Comment 2: Was antibiotic intake associated with gut microbiome?
Response 2: We analyzed and demonstrated the association of antepartum antibiotics use and their relations to gut microbiome β-diversity in Figure S3 (G). In the PCoA figure, Adonis P =0.36, indicates that antibiotics intake is not associated with gut microbiome.
Comment 3: Did maternal history of allergy affect any of your exposure or outcome data?
Response 3: We thank the reviewer for this comment. In Table R2 below, maternal history of allergy did not affect exposures and outcomes. We also consider whether father and parental history of allergy affect the exposures or outcomes, but both of them did not confound the relationship. In Figure R2, no significant difference was observed in alpha and beta diversity of gut microbiome among different maternal history of allergy.
Comment 4: Why did you ask about allergic rhinitis – very difficult to measure at one year.
Response 4: Thank you much for your concern. We agreed with your comment that allergic rhinitis is less prevalent at age one year. However, some documents still reported the prevalence of allergic rhinitis at infancy and young children stage was around 13.0-18.9% in China (Allergy 2018;73:1232–1243). Moreover, we used the globally used, validated ISAAC questionnaire to assess the physician-diagnosed allergic rhinitis. The allergic diseases determined questionnaire was adopted from ISSAC questionnaire, which has been validated and used to determine asthma cases throughout the world (Eur Respir J vol. 8, no. 3, pp. 483–91, 1995.). Furthermore, parental report of physician diagnosis has been found to accurately reflect physician diagnoses (Clin Exp Allergy vol. 22, no. 5, pp. 509–510, 1992.). The question we used to ask the mother was “Has a doctor diagnosed your child as having allergic rhinitis in the past 12 (or 2) months?” However, in our follow ups survey at age one year, more than half (7/13=0.54) of the mother responded “Yes” to this question.
Comment 5: What about food allergy – do you have any data?
Response 5: As suggested by the reviewer, we have assessed food allergy in this study. However, all parents reported their infants did not have the problem “food allergy”. On the other hand, their infants may suffer from other allergic problems such as respiratory tract allergic symptoms. In Table R3 below, other allergic problems did not affect exposures and outcomes.
Comment 6: You found no effect of diversity, the main driver of butyrate production. Do you think other fibers may be more important than fruit and vegetables to drive diversity?
Response 6: As suggested by the reviewer, we divided participants into low vs. high group according to mean of dietary fiber (13 g per day). The dietary fiber contains soluble and insoluble fibers. Most mothers consumed soluble fibers. In addition, we have a lack of information on insoluble fibers so we did not divide participants into two groups according to insoluble fibers. High or low consumption of dietary fiber during pregnancy did not affect both diversity of the infant’s gut microbiome.
Comment 7: Why would the 3 strains you found be association with atopic dermatitis outcomes? Do they have a specific role in the gut microbiome? Have others found these strains are important in atopic dermatitis prevention?
Response 7: The three strains (Erysipelatoclostridium, Betaproteobacteriales, and Lachnospiraceae) were identified from infants microbiome taxonomies from low maternal fruit and vegetable consumption group using LEfSe. Furthermore, nine strains were significantly associated with maternal nutritional intake (Figure 3) and were used to establish a prediction model of allergic diseases. Betaproteobacteriales and Erysipelatoclostridium had favorable predictive effects on the risk of atopic dermatitis at 2 months of age (Figure 4A). The CART-AD model of the entire data set demonstrated an accuracy of 82% and specificity of 85% but a low sensitivity of 38%. The CART-AD model of the validation data set yielded similar results. Betaproteobacteriales also presented a favorable predictive effect on the risk of atopic dermatitis at 1 year of age (Figure 4B) (entire data set: accuracy, 85%; specificity, 100%; and sensitivity, 60%). Moreover, Lachnospiraceae had a probable predictive effect on the risk of allergic rhinitis at 1 year of age (Figure 4C) (entire data set: accuracy, 85%; specificity, 67%; and sensitivity, 100%).
The three strains were evidenced to be linked to atopic diseases in the past literature. Erysipelatoclostridium was demonstrated to have a probable predictive effect on the risk of allergic diseases in southwest China during children’s first year of life (BMC Microbiology 2019: 19(1): 123). Betaproteobacteria was reported to have higher fraction among patients with atopy as compared with normal patients (Terapevticheskii Arkhiv; 2020; 92(3):56-60). Betaproteobacteria was indicated to be a dominant bacterial class associated with allergic diseases (PNAS 2012: 109(21): 8334-8339). A Taiwanese twin study revealed that Lachnospiraceae was significantly more common in children with allergic diseases than in those without allergic diseases from 2 to 12 months of age (Gastroenterology 2018: 154(1): 154-167). In another paper, the presence of Lachnospiraceae was associated with alteration of functional genes related to host immune development (J Allergy Clin Immunol
2018;141:1310-9). Moreover, Erysipelatoclostridium and Lachnospiraceae were evidenced to show higher richness among patients with allergic reactions as compared to controls (Int Immunopharyncology 2020; 84:106557). Taken together, the findings indicate that these three taxa, which were identified in the infants in the low fruit and vegetable consumption group, could be possible pathogens.

Reviewer 2 Report
Line 54: Suggest having some references here.
Line 57: Sentence not clear. This statement sounds like maternal nutrition only has negative influences on the child after birth. Is that true?
Line 61: Maybe it is good to specify what animal model was used in the previous study.
Line 71: Need citation for the first sentence.
Line78: It is good to specify whether it is maternal or infant gut microbiome.
Line 89: How did authors consider the maternal medical condition in the study? What is the exclusion criteria for the study?
Line 94: Is there any specific reason for the drastic reduce in number of babies providing a stool sample? Tedious sample collection? Cultural reason? This seems to be a limitation and can be addressed in the discussion for future studies.
Line 95: What is the rationale to collect the stool sample at 2-month? Will vary in terms of food/nutrients that babies consumed (eg: formula milk/breastmilk) until the stool sample collection affect the results? If not, why?
Line 103: Which trimester is the dietary record taken? Will it matter? Any particular reason that “three matching days” were used for the analysis?
Line 110: and along the manuscript: Is there any specific category of fruit and vegetables? eg starchy?
Line 124: The statement [DNA stabilization contained RNAlater, which can protect DNA from degradation at room temperature for days to weeks.] needs a citation.
Line 128: and along the manuscript: Are all the P-values reported in the study corrected for multiple testing? Eg: Benjamini Hochberg method?
Line 200: Table 1 From what I understand from this table, it seems like this study included only babies that were born at term, It is good that if the authors can specify this in the methodology section.
Line 202: It will be good if the resolution of Figure 1 be improved.
Line 232: Figure 2, need to include Adonis and betadisper values to be consistent with supplementary results.
Line 240: What statistical analysis was used to perform the correlation analysis?
Line 282: Authors reported that their findings have clinical implications for refining dietary recommendations in pregnancy to prevent infant gut dysbiosis, maybe need more detailed explanation here about “gut dysbiosis”.
Line 340: Regarding fructose, more explanation might be helpful. I might be overlooked, but in line 241, the authors are saying that A high-fructose and high-glucose maternal diet was significantly associated with an enrichment of Erysipelato-clostridium and Isobaculum, respectively but here the authors mentioned we highlight that maternal exposure to fructose reduces the abundance of Erysipelatoclostridium. The authors might also want to clarify the sentence on the role that fructose plays in affecting gut permeability.
Line 354: Need a collective reference here to support the statement
Supplementary documents: The number in Figure S1 does not match. Particularly on the first and the final stage.
Author Response
Dear Editor,
Enclosed please find our revised manuscript. We thank you for allowing us the opportunity to revise our manuscript. Our responses to the reviewers’ comments are attached in the following pages. We have carefully answered all concerns, and some alterations (highlighted in red font) have been made according to the reviewers’ suggestions. We have rewrite the introduction and methodology section to address the reviewer’s concern. We really appreciate the helpful comments and hope that our manuscript is now acceptable for publication in your prestigious journal.
Response to Reviewer # 2 comments
Comment 1: Line 54: Suggest having some references here.
Response 1: Thank you for your suggestion. We have added one reference (Nutrition reviews 2020;78(11):928-938) on line 54. In the abstract of this review article, the authors mentioned about plant based foods might protect against asthma development and improved asthma symptoms through their effects on systemic inflammation, oxidation and microbial composition. They have also discussed about the potential mechanisms linking high fruits and vegetables to protection of allergic diseases. Many mechanisms have been proposed, but few have been proven.
Comment 2: Line 57: Sentence not clear. This statement sounds like maternal nutrition only has negative influences on the child after birth. Is that true?
Response 2: We should apologize for the unclear sentence. What we like to state is that gestational nutrition may influence offspring physiology and susceptibility to disease through alterations in infant gut microbiome. Hence, we rephrased our sentence to “Maternal nutrition during pregnancy may affect mother-to-child transmission of bacteria, resulting in changes of gut microflora in the child, with long-term consequences after birth” on line 57.
Comment 3: Line 61: Maybe it is good to specify what animal model was used in the previous study.
Response 3: Following your suggestion, we specify the detailed animal model used in the previous studies. In the Chu and Ma et al’s paper, they undertook microbiome metagenomics studies by using their well-characterized, outbred Japanese macaque (Macaca fuscata) model of maternal obesity. In the Li et al’s paper, by feeding the sow with different components of fiber during gestation, they discovered that the composition of dietary fiber in pregnancy diet had an essential role in improving antioxidant capacity and decreasing inflammatory response of mothers and their offspring through gut microbiome composition modulation. We modified our manuscript as follows, “Chu et al. discovered that a high-fat maternal diet during gestation shapes the offspring gut microbiome in animals (Japanese Macaque)[6] and humans [7]. Using a sow model, Li et al. reported that maternal dietary fiber intake alters offspring gut microbiome composition [8].”
Comment 4: Line 71: Need citation for the first sentence.
Response 4: Thank you for your suggestion. We now cited a reference for the first sentence (J Allergy Clin Immunol. 2017; 139(4): 1099–1110). There has been strong evidence that early infancy microbiome plays an important role in modulating the early immunologic functions. The citing reference was listed as below, “Early infancy gut microbiome predicts future allergic diseases [10].”
Comment 5: Line78: It is good to specify whether it is maternal or infant gut microbiome.
Response 5: Yes, we have specified on line 78 that we meant infant gut microbiome. Our manuscript was revised as follows, “(2) investigated the interrelationships between maternal nutrients and abundance of infants’ gut microbiome taxa”
Comment 6: Line 89: How did authors consider the maternal medical condition in the study? What is the exclusion criteria for the study?
Response 6: As suggested by the reviewer, the detail of exclusion criteria was described in our methods as the follow: “Pregnant women with severe diseases (e.g., heart diseases) were also excluded. Initially we included 479 participants who completed the baseline survey and provided dietary data, and 199 of them participated in newborn follow-up visits. We excluded participants who were un-willing to provide infant stool samples. By the end of April 2020, 39 infant stool samples were obtained at 2 months postpartum.” We have further described the maternal medical condition in Table 1. We observed that there is no significant difference in the proportion of maternal diseases between high and low consumption of vegetables and fruits.
Comment 7: Line 94: Is there any specific reason for the drastic reduce in number of babies providing a stool sample? Tedious sample collection? Cultural reason? This seems to be a limitation and can be addressed in the discussion for future studies.
Response 7: We pointed out in the limitation section that the generalizability of our findings may be limited by our relatively small sample size (Line 382). We originally expected that 1/3 of the mothers interviewed during pregnancy can accept home visit follow-ups survey. However, the majority of mother need to come back working after two months they gave birth to their offspring, making collecting of the stool sample and receiving home visit difficult. Moreover, due to the start of COVID-19 pandemic in early 2020, a large proportion of mother feels hesitate to accept any visit from the hospital, making the collection of stool sample even more difficult.
Comment 8: Line 95: What is the rationale to collect the stool sample at 2-month? Will vary in terms of food/nutrients that babies consumed (eg: formula milk/breastmilk) until the stool sample collection affect the results? If not, why?
Response 8: We aimed to explore the impact of high/low gestational intake of fruits and vegetables on infant microbiome and related taxa in this manuscript. According to Chu et al’s paper (Genome Medicine 2016;8:77), they collected the infant’s stool sample at 6 weeks of age and reported that a maternal high-fat diet is associated with distinct changes in the neonatal gut microbiome which persist through 4-6 weeks after birth. Ma et al’s primate model (Nat Commun. 2014; 5: 3889) has provided a high-fat diet or a control diet prior and during pregnancy and lactation. Then the offspring were weaned onto a control diet at 6-7 months of age. Surprisingly, at 1 year of age, the offspring gut microbiome could be discriminated based on whether their mothers consumed a high-fat or control-diet, despite the offspring consume a control diet for several months. To avoid the influence of solid food introduction (usually starting from 4-6 months), and to consider the convenience for mothers to cooperate in collection the stool sample, we decided to perform the home visit at 2 months postpartum to collect the infant stool sample.
Comment 9: Line 103: Any particular reason that “three matching days” were used for the analysis?
Response 9: The majority of the dietary record were taken in 2nd and 3rd trimester. The three matching days (containing one weekday and two weekend days) of dietary records were used for analysis. Mean of three-days dietary record were considered reliable to avoid day-to-day variations in assessing dietary intake. We previously proved the validity and reliability of this three-days image-based dietary assessment application for assessing macronutrients and micronutrients (Nutrients. 2019 Oct 20;11(10). pii: E2527).
Comment 10: Line 110: and along the manuscript: Is there any specific category of fruit and vegetables? eg starchy?
Response 10: There is not a specific category of fruit and vegetables. In Taiwan, most women consumed 4.9 g/day of dietary fiber, well below the recommended 25 g/day. Most dietary fibers were obtained from vegetables and fruits so the mean cups of fruits and vegetables were estimated as follows: five cups of fruits and vegetables (the minimum recommended) multi-plied by 20% (derived from the average 4.9 g/day divided by the recommended 25 g/day of dietary fiber). We then determined The high or low consumption of vegetables and fruit based on more than one cup of fruits and more than one cup of vegetables per day. In addition, we have a lack of information on starchy so we did not divide participants into two groups according to starchy or specific vegetables and fruits.
Comment 11: Line 124: The statement [DNA stabilization contained RNAlater, which can protect DNA from degradation at room temperature for days to weeks.] needs a citation.
Response 11: We have added the reference (PeerJ, 7, e8133) in the revised manuscript.
Carruthers, L. V., Moses, A., Adriko, M., Faust, C. L., Tukahebwa, E. M., Hall, L. J., Ranford-Cartwright, L. C., & Lamberton, P. (2019). The impact of storage conditions on human stool 16S rRNA microbiome composition and diversity. PeerJ, 7, e8133.
Comment 12: Line 128: and along the manuscript: Are all the P-values reported in the study corrected for multiple testing? Eg: Benjamini Hochberg method?
Response 12: Yes, all the P-values reported in the study were corrected for multiple testing by Bonferroni post hoc test. We also stated this point in the section of 2.4 Targeted 16S rRNA gene sequencing, “One-way ANOVA followed by Bonferroni post hoc test were provided to handle multiple comparison analysis” on line 148.
Comment 13: Line 200: Table 1 From what I understand from this table, it seems like this study included only babies that were born at term, It is good that if the authors can specify this in the methodology section.
Response 13: Thank for this comment as it points to an important rationale of this study. We have replaced the sentence as “Other potential confounders, such as gestational age (preterm birth or not), excess gestational weight gain, mode of delivery, and breastfeeding or formula feeding, were also not significantly different (all p > 0.05).(Line 198)”
Comment 14: Line 202: It will be good if the resolution of Figure 1 be improved.
Response 14: As suggested by the reviewer, we re-made a high-quality illustration as below (Figure 3).
Comment 15: Line 232: Figure 2, need to include Adonis and betadisper values to be consistent with supplementary results.
Response 15: As suggested by the reviewer, we included Adonis and betadisper values to be consistent with supplementary results (Line 238).
Comment 16: Line 240: What statistical analysis was used to perform the correlation analysis?
Response 16: We apologize for the unclarified descriptions. As suggested by the reviewer, we added a sentence in the statistical method as “Chi-square and t test were used to examine whether demographic characteristics and maternal nutrients differ between the groups. Linear regression was used to examine the relationship between nutrients and the infant gut microbiome at 2 months of age” (Line 173).
Comment 17: Line 282: Authors reported that their findings have clinical implications for refining dietary recommendations in pregnancy to prevent infant gut dysbiosis, maybe need more detailed explanation here about “gut dysbiosis”.
Response 17: We have revised the sentence in the revised manuscript. As follows (Line 290): “These findings have clinical implications for refining dietary recommendations in pregnancy to prevent infant gut dysbiosis, associated with high abundance of Erysi-pelatoclostridium, Betaproteobacteria, and Lachnospiraceae, and early-life allergic diseases.”
Comment 18: Line 340: Regarding fructose, more explanation might be helpful. I might be overlooked, but in line 243, the authors are saying that A high-fructose and high-glucose maternal diet was significantly associated with an enrichment of Erysipelato-clostridium and Isobaculum, respectively but here the authors mentioned we highlight that maternal exposure to fructose reduces the abundance of Erysipelatoclostridium. The authors might also want to clarify the sentence on the role that fructose plays in affecting gut permeability.
Response 18: We apologize for the misleading and thank the reviewer for pointing this out. In the results, we corrected the statement as “A high-fructose maternal diet was negatively associated with Erysipelatoclostridium. A high-glucose maternal diet was significantly associated with an enrichment Isobaculum.” (Line 247).
Comment 19: Line 354: Need a collective reference here to support the statement.
Response 19: Thank you for your advice. We provided a collective reference to support this statement, as follows: “In the present study, Erysipelatoclostridium, Betaproteobacteria, and Lachnospiraceae were the specific intestinal microbes linked to allergic diseases, consistent with previous studies [43-45]” (Line 363).
Comment 20: Supplementary documents: The number in Figure S1 does not match. Particularly on the first and the final stage.
Response 20: We apologize for the misleading and thank the reviewer for pointing this out. As suggested by the reviewer, we corrected the number in the flowchart of participant recruitment.

Reviewer 3 Report
I have many concerns with this manuscript. Firstly, the abstract implies that this study involved 199 women, but actually it is based on only 39 mother-infants who had stool samples collected at 2 months of age and only 13 children were followed up for allergic disease outcomes at 1 year of age. Although atopic dermatitis is an appropriate outcome at 1 year of age, allergic rhinitis is not (as should be rare at 1 year of age). In the supplementary tables I found that only 5 infants had atopic dermatitis at 1 year thus this is insufficient case numbers to base any results on. The predictive model based on only 39 infant stool samples and only 5 cases of atopic dermatitis at 1 year is very likely to be biased and should be reported clearly as an exploratory pilot analysis approach at best or even ideally these allergic disease outcomes with clearly insufficient number of children that were followed up should all be removed from the manuscript.
In addition, the method used to determine low and high fruit and vegetable consumption groups was unusual and resulted in only 13 women in the high consumption group compared to 26 in the low group. A quartile approach would seem more appropriate but the n=39 is simply too small.
My other minor wording suggestions for enhancement of this manuscript introduction are as follows (discussion wording recommendations are not being made as the methods and results have too many flaws):
Lines 49-50: Please reword or even delete the first sentence as increased fruit and vegetable consumption has been associated with the reduced development of allergic diseases (as per second sentence) but not yet proven in randomized controlled trials to ‘protect’ against allergic disease.
Line 58: Please reword ‘resulting in poor gut microflora in the child’ to indicate a gut microflora profile that may be associated with poorer health outcomes in the child.
Line 65: replace ‘microbiome group’ with microbiome profile or microbiome composition
Line 71: The sentence “Early infancy gut microbiome predicts future allergic diseases” needs to be referenced.
Author Response
Dear Editor,
Enclosed please find our revised manuscript. We thank you for allowing us the opportunity to revise our manuscript. Our responses to the reviewers’ comments are attached in the following pages. We have carefully answered all concerns, and some alterations (highlighted in red font) have been made according to the reviewers’ suggestions. We have rewrite the introduction and methodology section to address the reviewer’s concern. We really appreciate the helpful comments and hope that our manuscript is now acceptable for publication in your prestigious journal.
With very best wishes and warm regards.
Response to Reviewer # 3 comments
Comment 1: I have many concerns with this manuscript. Firstly, the abstract implies that this study involved 199 women, but actually it is based on only 39 mother-infants who had stool samples collected at 2 months of age and only 13 children were followed up for allergic disease outcomes at 1 year of age. Although atopic dermatitis is an appropriate outcome at 1 year of age, allergic rhinitis is not (as should be rare at 1 year of age). In the supplementary tables I found that only 5 infants had atopic dermatitis at 1 year thus this is insufficient case numbers to base any results on. The predictive model based on only 39 infant stool samples and only 5 cases of atopic dermatitis at 1 year is very likely to be biased and should be reported clearly as an exploratory pilot analysis approach at best or even ideally these allergic disease outcomes with clearly insufficient number of children that were followed up should all be removed from the manuscript.
Response 1: We apologize for the misleading and thank the reviewer for pointing this out. As suggested by the reviewer, we clarify the study design as “This exploratory pilot study is to explore the effect of gestational intake of high or low vegetable and fruit intake on infant microbiome and its relationship to allergic diseases at 1 year of age” in the abstract.
The three strains (Erysipelatoclostridium, Betaproteobacteriales, and Lachnospiraceae) were identified from infants microbiome taxonomies from low maternal fruit and vegetable consumption group using LEfSe. The three strains were evidenced to be linked to atopic diseases in the past literature. Erysipelatoclostridium was demonstrated to have a probable predictive effect on the risk of allergic diseases in southwest China during children’s first year of life (BMC Microbiology 2019: 19(1): 123). Betaproteobacteria was reported to have higher fraction among patients with atopy as compared with normal patients (Terapevticheskii Arkhiv; 2020; 92(3):56-60). Betaproteobacteria was indicated to be a dominant bacterial class associated with allergic diseases (PNAS 2012: 109(21): 8334-8339). A Taiwanese twin study revealed that Lachnospiraceae was significantly more common in children with allergic diseases than in those without allergic diseases from 2 to 12 months of age (Gastroenterology 2018: 154(1): 154-167). In another paper, the presence of Lachnospiraceae was associated with alteration of functional genes related to host immune development (J Allergy Clin Immunol 2018;141:1310-9). Taken together, the findings indicate that these three taxa, which were identified in the infants in the low fruit and vegetable consumption group, could be possible pathogens.
Comment 2: In addition, the method used to determine low and high fruit and vegetable consumption groups was unusual and resulted in only 13 women in the high consumption group compared to 26 in the low group. A quartile approach would seem more appropriate but the n=39 is simply too small.
Response 2: Yes, our sample size is a bit small so that the quartile approach is not suitable for grouping participants. Actually, our method based on Taiwan National Nutrition and Health Survey is more practicable. In Taiwan, most women consumed 4.9 g/day of dietary fiber, well below the recommended 25 g/day. The mean cups of fruits and vegetables was estimated as follows: five cups of fruits and vegetables (the minimum recommended) multi-plied by 20% (derived from the average 4.9 g/day divided by the recommended 25 g/day of dietary fiber). We then determined The high or low consumption of vegetables and fruit based on more than one cup of fruits and more than one cup of vegetables per day.
Comment 3: Lines 49-50: Please reword or even delete the first sentence as increased fruit and vegetable consumption has been associated with the reduced development of allergic diseases (as per second sentence) but not yet proven in randomized controlled trials to ‘protect’ against allergic disease.
Response 3: Yes, we agreed with your comments. We have now deleted the first sentence on line 49-50.
Comment 4: Line 58: Please reword ‘resulting in poor gut microflora in the child’ to indicate a gut microflora profile that may be associated with poorer health outcomes in the child.
Response 4: We agreed with your comment. We have now amended our description to “Maternal nutrition during pregnancy may affect mother-to-child transmission of bacteria, resulting in changes of gut microflora in the child, with long-term consequences after birth [5].”
Comment 5: Line 65: replace ‘microbiome group’ with microbiome profile or microbiome composition.
Response 5: Following your suggestion, we have used “microbiome composition” instead of microbiome group. The sentence was then described as follows, “Similarly, maternal fruit intake was associated with increased risk of a high Streptococcus/Clostridium gut microbiome composition among vaginally delivered infants [9].”
Comment 6: Line 71: The sentence “Early infancy gut microbiome predicts future allergic diseases” needs to be referenced.
Response 6: Thank you for your suggestion. We now cited a reference for the first sentence (J Allergy Clin Immunol. 2017; 139(4): 1099–1110). There has been strong evidence that early infancy microbiome plays an important role in modulating the early immunologic functions. The citing reference was listed as below, “Early infancy gut microbiome predicts future allergic diseases [10].”

Round 2
Reviewer 1 Report
Thank you - my concerns have been addressed.
Author Response
Enclosed please find our revised manuscript. We thank you for allowing us the opportunity to revise our manuscript. Our responses to the reviewers’ comments are attached in the following pages. We have carefully answered all concerns, and some alterations (highlighted in red font) have been made according to the reviewer's suggestions. We have removed the outcome “allergic diseases” from the entire manuscript. We also re-wrote the introduction, methods, and discussion section to address the reviewer’s concern. We really appreciate the helpful comments and hope that our manuscript is now acceptable for publication in your prestigious journal.
With very best wishes and warm regards.

Reviewer 3 Report
This manuscript remains misleading and still requires major amendments. The abstract remains misleading as it implies that this study involved 199 women, but only 13 children were followed up for allergic disease outcomes at 1 year of age. Table 1 on demographic characteristics only includes the 39 women with matching stool samples. These details need to be in the abstract. Given that only 13 infants were followed-up at 1 year this is a poor quality follow-up and hence all inclusion of the allergic disease outcomes should be removed. As previously mentioned the predictive model based on only 39 infant stool samples and only 5 cases of atopic dermatitis at 1 year is very likely to be biased and should be removed from the manuscript. The allergic disease outcomes should be removed from the title, methods and results.
Author Response
Enclosed please find our revised manuscript. We thank you for allowing us the opportunity to revise our manuscript. Our responses to the reviewers’ comments are attached in the following pages. We have carefully answered all concerns, and some alterations (highlighted in red font) have been made according to the reviewer's suggestions. We have removed the outcome “allergic diseases” from the entire manuscript. We also re-wrote the introduction, methods, and discussion section to address the reviewer’s concern. We really appreciate the helpful comments and hope that our manuscript is now acceptable for publication in your prestigious journal.
With very best wishes and warm regards.
Response to Reviewer # 3 comments
Comment 1: This manuscript remains misleading and still requires major amendments. The abstract remains misleading as it implies that this study involved 199 women, but only 13 children were followed up for allergic disease outcomes at 1 year of age. Table 1 on demographic characteristics only includes the 39 women with matching stool samples. These details need to be in the abstract. Given that only 13 infants were followed-up at 1 year this is a poor quality follow-up and hence all inclusion of the allergic disease outcomes should be removed. As previously mentioned the predictive model based on only 39 infant stool samples and only 5 cases of atopic dermatitis at 1 year is very likely to be biased and should be removed from the manuscript. The allergic disease outcomes should be removed from the title, methods and results.
Response 1: We should apologize for the misleading so we removed the outcome “allergic diseases” from the entire manuscript. In the abstracts (Line 27 to 40), we re-wrote the background and highlighted the sample size "n=39". We removed "allergic diseases” from the results. In the introduction section, we re-write background, motivation, and purpose (Line 47 to 81). Finally, we added a section regarding maternal vitamin intake and gut microbiome (Line 292 to 297).
Abstracts (Line 27 to 40)
Abstract: Maternal nutrition during pregnancy may affect mother-to-child transmission of bacteria, resulting in gut microflora changes in the offspring, with long-term health consequences in later life. Only a small amount of evidence showing the effect of nutrition intake during pregnancy on the gut microbiome of infants and these studies have been majorly conducted in animal models. Hence, longitudinal human studies are lacking. This exploratory pilot study explores the effect of gestational intake of high or low vegetable and fruit intake on infant microbiome. We enrolled pregnant women with a complete 3-day dietary record and received postpartum follow-up. The 16S rRNA gene sequence was used to characterize the infant gut microbiome at 2 months (n=39). Principal coordinate analysis ordination revealed that the infant gut microbiome clustered differently for high or low maternal fruit and vegetable consumption (p < 0.001). The linear discriminant analysis effect size feature selection identified 6 and 17 taxa from the high and low fruit and vegetable consumption groups. Among the 23 abundant taxa, we observed maternal 6 nutrients were associated with 9 taxa (e.g., Erysipelatoclostridium, Isobaculum, Lachnospiraceae, Betaproteobacteriales, Burkholderiaceae, Sutterella, Clostridia, Clostridiales, and Lachnospiraceae). The amount of gestational vegetable and fruit consumption is associated with distinct changes in the infant gut microbiome at 2 months of age. Therefore, strategies involving increased fruit and vegetable consumption during pregnancy should be employed for modifying the gut microbiome early in life.
Introduction (Line 47 to 81)
According to the Development Origins of Health and Disease (DOHaD) Hypothesis, maternal nutrition in pregnancy has a significant impact on offspring disease risk in the future [1]. A maternal diet rich in fruits and vegetables during pregnancy is associated with a reduced risk of allergic diseases [2], and an increased risk of obesity [3]. Given that nutrients intake strongly influence microbiome function and relative abundance, the infant gut microbiome might be a potential mediating factor linking gestational nutritional exposure and future childhood diseases [4].
Maternal nutrition during pregnancy may affect mother-to-child transmission of bacteria, resulting in gut microflora changes in the child, with long-term consequences after birth [4]. However, evidence supporting the effect of maternal nutrition during pregnancy on the infant gut microbiome remains scarce, and most relevant studies have been conducted on animal models. Chu et al. discovered that a high-fat maternal diet during gestation shapes the offspring gut microbiome in animals (Japanese Macaque)[5] and humans [6]. The mother's high-fat diet has been shown to damage the microbiome and immune system of their offspring [7]. The offspring of mothers who consume Western diets have significantly increased the effect of Pachycephalospora on Bacteroides, and the microbiome of the offspring of mothers who fed a high-fat diet increased their ability to extract energy from the diet. Using a sow model, Li et al. reported that maternal dietary fiber intake alters offspring gut microbiome composition [8]. Similarly, maternal fruit intake was associated with an increased risk of a high Streptococcus/Clostridium gut microbiome composition among vaginally delivered infants [9]. Possible mechanisms for the effects of maternal diet during pregnancy on the infant gut microbiome include the transmission of nutrients through amniotic fluid, vaginal delivery, or the placenta. However, the effect of gestational intake of high or low vegetable and fruit intake on infant microbiome remains unclear in the human study.
Several studies suggest that supplementation with nutrients rich in vegetables and fruits, such as dietary fiber, Vitamin C, and fructose, could modulate the structure of host gut microbes [10]. According to a previous study by Alison et al., a high-fiber diet alters gut microbial ecology and causes significant perturbations at the phylum level [11]. Li et al. informed that vitamin C could strongly modulate the gut microbiota [12]. In another animal study, the maternal diet supplemented with fructose appeared to regulate the maternal microbiome significantly, causing infant gut dysbiosis [13]. Besides, the effect of maternal dietary vegetables and fruits may not only on the host but also on their next-generation [14, 15]. But to date, this relationship has still been poorly understood.
In this study, we (1) explored the impact of high/low gestational intake of fruits and vegetables on infant microbiome and (2) investigated the interrelationships between maternal nutrients and abundance of infant gut microbiome taxa.
Discussion (Line 292 to 297)
However, the effects of micronutrients in the above association in the mother or child remain unclear. Regarding maternal vitamin intake and gut microbiome, the intake of ascorbic acid (vitamin C) during pregnancy was positively correlated with the abundance of Staphylococcus [45]. Although the role of ascorbic acid in Staphylococcus metabolism remains unclear, both have been linked to the immune profile [46]. In our study, we found the effect of ascorbic acid on Betaproteobacteriales, Burkholderiaceae. The role of ascorbic acid on Betaproteobacteriales and Burkholderiaceae metabolism is still unclear.
